# Predictors of injury mortality: findings from a large national cohort in Thailand

Vasoontara Yiengprugsawan,[1] Janneke Berecki-Gisolf,[2] Christopher Bain,[1,3] Roderick McClure,[2] Sam-ang Seubsman,[1,4] Adrian C Sleigh[1]

## ABSTRACT

**Objective:** To present predictors of injury mortality by types of injury and by pre-existing attributes or other individual exposures identified at baseline.

**Design:** 5-year prospective longitudinal study.

**Setting:** Contemporary Thailand (2005–2010), a country undergoing epidemiological transition.

**Participants:** Data derived from a research cohort of 87 037 distance-learning students enrolled at Sukhothai Thammathirat Open University residing nationwide.

**Measures:** Cohort members completed a comprehensive baseline mail-out questionnaire in 2005 reporting geodemographic, behavioural, health and injury data. These responses were matched with national death records using the Thai Citizen ID number. Age–sex adjusted multinomial logistic regression was used to calculate ORs linking exposure variables collected at baseline to injury deaths over the next 5 years.

**Results:** Statistically significant predictors of injury mortality were being male (adjustedOR 3.87, 95% CI 2.39 to 6.26), residing in the southern areas (AOR 1.71, 95% CI 1.05 to 2.79), being a current smoker (1.56, 95% CI 1.03 to 2.37), history of drunk driving (AOR 1.49, 95% CI 1.01 to 2.20) and ever having been diagnosed for depression (AOR 1.91, 95% CI 1.00 to 3.69). Other covariates such as being young, having low social support and reporting road injury in the past year at baseline had moderately predictive AORs ranging from 1.4 to 1.6 but were not statistically significant.

**Conclusions:** We complemented national death registration with longitudinal data on individual, social and health attributes. This information is invaluable in yielding insight into certain risk traits such as being a young male, history of drunk driving and history of depression. Such information could be used to inform injury prevention policies and strategies.

For numbered affiliations see end of article.

**Correspondence to**
Dr Vasoontara Yiengprugsawan;
vasoontara.yieng@anu.edu.au,
vasoontara.yieng@gmail.com

## Strengths and limitations of this study

- Injury is a population health burden in transitional low-income and middle-income Southeast Asia. We investigated a large national cohort of Thai adults for predictors of injury mortality including geodemographic, social and health attributes recorded at baseline.
- Injuries constituted almost one-third of all deaths in the cohort, and some 40% of those were from transport and nearly 60% were non-transport injuries. These injury mortality observations add to our previous Thai work on injury morbidity, highlighting the overall risks, especially depression, male sex and drunk driving.
- The advantage of our study is its large size, longitudinal design and comprehensive baseline information. This provides a platform for identification of risks, elimination of confounders and exploration of causal pathways.
- This study captured the 5-year mortality rate in a generally young adult cohort. Thus there were relatively few deaths. Citizen IDs provided at baseline will enable us to monitor patterns of cohort mortality into the future.

1.3 million deaths and many suffered from serious physical and mental consequences.[5][6]

Most national injury prevention policies have been introduced in high-income countries. Unfortunately, very few low-income and middle-income countries have been able to develop such policies due to lack of resources and limited availability of quality injury mortality data.[7][8] In particular, many developing countries still face the challenges of accurately identifying causes of death from routinely collected national civil registration and vital statistics systems while other sources of data, such as police reports and hospital records, are never comprehensive and lead to under-reporting bias if relied on as the main source of injury mortality data. Reliable cause-of-death data are important because they enable monitoring of the epidemiological occurrence and public health effects at the population level.[9][10]

## INTRODUCTION

Injury remains a major public health challenge worldwide, causing one-tenth of global mortality with a heavy burden in developing countries.[1][2] According to the WHO, at least 1.2 million people are killed from road crashes and an estimated 50 million are injured on roads worldwide each year.[3][4] Violence and non-transport injuries also accounted for more than

Throughout middle-income Southeast Asia, including Thailand, injury continues to be one of the top 10 causes of death.[11][12] In past decades, Thailand has reformed administrative records to improve the coverage and quality of cause-specific mortality data.[13][14] Eight years ago, Thailand began to study the ill-defined causes of death by using verbal autopsies and these revealed that besides a high proportion of transport-related deaths, a number of other deaths which were initially recorded as non-specific causes turned out to be suicides, assaults and drowning.[15–17] These findings shed light on the importance of non-transport injuries in addition to the burden of transport injuries.

This study is based on a large national cohort in Thailand which has been followed to investigate health-risk transitions of Thai adults since 2005. The cohort database includes comprehensive information on individual characteristics, social demography, health behaviours and specific diseases, as well as history of injuries. Our previous research based on this cohort has examined risk factors associated with injury morbidity.[18–20] Now successful mortality data linkage through the Thai Ministry of Interior and Ministry of Public Health allows us to analyse injury-related deaths among the cohort over the first 5 years (2005–2010).

Informed by our earlier research on injury morbidity, and by related published information, this study has investigated injury in more depth using mortality as the outcome. Our study linked cohort outcomes (survival, non-injury death and injury death) to an array of relevant exposures recorded at baseline including geodemographic attributes, social covariates, health and psychological states and health-risk behaviours. This analysis is prospective and cohort-based and fills an important gap regarding our knowledge of injury risks as an emerging public health problem in a middle-income Asian country going through the health-risk transition.

## METHODS
### Study population and data collection
This analysis is part of the overarching Thai Cohort Study (TCS), an ongoing epidemiological investigation of changing patterns of health risks and outcomes. Data are derived from a research cohort of 87 037 distance-learning adult students enrolled at Sukhothai Thammathirat Open University, who resided all over Thailand and completed the baseline comprehensive mail-out health questionnaire in 2005 (response rate 44%). The cohort participants recapitulated well the distance-learning student body at STOU and share certain geodemographic attributes with the general Thai population (mean age was 29 years in 2005, slightly more than half were women, half resided in urban areas).[21][22] The baseline questionnaire gathered data on a wide range of topics including age, sex, income, marital status, health status, doctor-diagnosed diseases, health-risk behaviours including smoking and drinking, social capital and history of injury.

### Mortality data
The completeness of death registration in Thailand was 86% from 1950 to 2000[10] but over the past decade coverage improved to 95%.[23] A powerful feature of our study is that all cohort members have provided their Thai Citizen ID number enabling detection and analysis of deaths in the future. These confidential ID numbers were safeguarded and stored at STOU in a secure office on the main campus with 24 h guards on patrol. The working files of these data were de-identified and no individual information will be released or displayed in any format. To detect deaths, the Bangkok TCS team periodically matched the cohort against national death records from the Ministry of Interior using the Citizen ID number. At a later stage the Thai Ministry of Public Health expanded these death records by adding the standard International Classification of Diseases (ICD-10)[24] to identify causes of death.

Up until March 2010, there were a total of 580 deaths among the TCS participants. According to the ICD-10 codes, there were 376 deaths from non-injury causes including ill-defined causes of death. For the purpose of this study, these will not be broken down and will be designated as 'other deaths'. For our injury-focused death analysis, there were 204 deaths from external causes, including 84 deaths from transport accidents. Among the 120 non-transport injury deaths, there were 35 deaths from miscellaneous external causes, 10 deaths from intentional self-harm, 30 deaths from assault and 45 deaths from 'unspecified events of undetermined intent'.

### Exposures and confounders
In our analysis, exposures of interest and potential confounders from the 2005 baseline questionnaire included the following geodemographic variables: age (4 categories), sex, marital status (married, not married, divorced/widowed), personal monthly income (≤3000 Baht, 3001–7000, 7001–10 000, 10 001–20 000, >20 000), regions (central/east, Bangkok, north, northeast, south) and lifecourse urbanisation (residence at age 12 years old and at baseline: rural–rural, rural–urban, urban–rural, urban–urban). As well, a history of injury in the past year was reported at 2005 baseline, including the frequency and location of injuries reported.

Also analysed as exposures of interest were certain social covariates, several health states and important health-risk behaviours. These behaviours included smoking and alcohol drinking which have been shown to be independently associated with injury.[25][26] Smoking status includes never, current and former and alcohol status includes never, occasional, regular and former. In addition, at baseline cohort members were asked 'in the last year have you driven a motor vehicle after consuming 3 or more glasses of alcohol' (ie, drunk driving). Other health-related attributes included self-assessed health and

chronic metabolic or cardiovascular disorders (eg, diabetes, hypertension). A history of doctor-diagnosed depression has previously been shown to be an injury risk[18 27] and is also included in the model. Social capital was dichotomoised for analyses (low or not low) in three domains: trust (whether people can be trusted), support (from family, friends, colleagues) and interaction (with family, friends, neighbours).

## Data processing and statistical analysis

Questionnaire responses were digitised by optical scanning and subsequently edited using Thai Scandevet, SQL and SPSS software. For analysis we used Stata V.12. Individuals with missing data for given analyses were excluded (<5%), so totals vary slightly according to available information. We described the distribution of

deaths by demographic, social and health covariates, by cause of death and by types of injury (transport and non-transport injuries). In addition, we presented death rates per 10 000 person-years of exposure. We then used age–sex-adjusted multinomial logistic regression linking the above covariates to three possible outcomes: alive (reference), deaths from other causes and deaths from injury (study outcome). The final analytical model mutually adjusted for all covariates and reported adjusted ORs (AOR) and 95% CIs.

## RESULTS

Among Thai cohort members at baseline in 2005 (table 1), about one-third of the cohort were less than 30 years old, slightly more than half were women, about 40% reported monthly income of less than 7000 Baht per month (US

**Table 1** Distribution of mortality by baseline social and geodemographic attributes, Thai Cohort Study

| Cohort attributes | Vital status by attributes, per cent | | | | | Incidence/10 000 person-years | |
| | Alive (86 457) | Other deaths (376) | Injury deaths (204) | Injury deaths (204) | | | |
| | | | | Transport (84) | Non-transport (120) | Transport (84) | Non-transport (120) |
|---|---|---|---|---|---|---|---|
| Age groups in years | | | | | | | |
| 18–29 | 31.4 | 27.7 | 27.9 | 20.2 | 25.7 | 6 | 15 |
| 30–39 | 53.6 | 29.5 | 59.3 | 69.1 | 51.4 | 12 | 14 |
| 40–49 | 12.5 | 25.8 | 9.3 | 6.0 | 17.1 | 5 | 13 |
| ≥50 | 2.4 | 17.0 | 3.4 | 4.8 | 5.7 | 19 | 14 |
| Sex | | | | | | | |
| Males | 45.2 | 65.6 | 73.7 | 69.4 | 76.7 | 15 | 23 |
| Marital status | | | | | | | |
| Married | 38.8 | 50.1 | 35.0 | 30.9 | 37.7 | 8 | 14 |
| Not married | 56.8 | 41.9 | 60.1 | 64.3 | 57.1 | 11 | 14 |
| | 4.4 | 8.0 | 4.9 | 4.8 | 5.0 | 11 | 16 |
| Personal monthly income | | | | | | | |
| ≤3000 Baht | 11.0 | 14.9 | 14.6 | 11.9 | 16.5 | 10 | 20 |
| 3001–7000 | 30.9 | 22.4 | 33.2 | 39.3 | 28.7 | 13 | 13 |
| 7001–10 000 | 23.3 | 16.6 | 23.6 | 26.2 | 21.7 | 11 | 13 |
| 10 001–20 000 | 24.2 | 28.7 | 20.6 | 16.7 | 23.5 | 7 | 13 |
| >20 000 | 10.5 | 17.4 | 8.0 | 6.0 | 9.6 | 6 | 12 |
| Regions | | | | | | | |
| Central/east | 30.7 | 30.7 | 24.9 | 23.8 | 24.2 | 8 | 11 |
| Bangkok | 17.2 | 18.7 | 10.7 | 9.4 | 11.7 | 5 | 9 |
| North | 18.2 | 22.7 | 21.0 | 23.5 | 19.2 | 13 | 15 |
| Northeast | 20.9 | 21.6 | 23.4 | 25.9 | 21.7 | 12 | 14 |
| South | 13.0 | 12.3 | 20.6 | 16.7 | 23.3 | 12 | 25 |
| Lifecourse residence | | | | | | | |
| Rural–rural | 43.3 | 42.3 | 46.6 | 38.1 | 52.5 | 8 | 17 |
| Rural–urban | 31.5 | 25.0 | 16.7 | 28.6 | 30.8 | 9 | 13 |
| Urban–rural | 4.2 | 6.9 | 4.4 | 7.1 | 2.5 | 16 | 8 |
| Urban–urban | 19.7 | 22.3 | 16.7 | 22.6 | 12.5 | 11 | 9 |

$175 in 2005), 18% lived in Bangkok and about half were urban residents. Injury deaths were more likely to affect men (73.7%, 15 vs 23 per 10 000 person-years for transport vs non-transport injuries). Also notable, injury deaths were disproportionately frequent in the southern region (20.6%, 12 vs 25 per 10 000 person-years for transport vs non-transport injuries).

For social and health attributes (table 2), a history of ever drunk driving in the past year was more common among injury deaths (42.4% compared with 26.5% for other deaths or 25.4% for alive) and notably higher for transport than non-transport injuries (52.9% vs 35%). Cohort members who died from non-injury-related causes were twice as likely to have reported poor self-assessed health at baseline and three times as likely to have reported metabolic and cardiovascular chronic conditions. Cohort members who died from injury reported higher rates of ever having doctor-diagnosed depression (6.9% compared with 3.4% among non-deaths). As well,

a history of depression was much more frequent for those who died from non-transport injuries than for transport injuries (34 vs 13/ per 10 000 person-years). At baseline in 2005, about 20% of cohort members overall reported injury at least once in the past year compared with 33.3% of cohort participants who died from transport injury.

In addition to analysing by injury types, we also tabulated the death rates according to the ICD (table 3). Within transport injury mortality, rates per 10 000 person-years for motorcycle riders and car occupants were 1.6 and 1.7, respectively. There were also 4.5/ 10 000 person-years who died in unspecified motor vehicles. Among non-transport injury deaths, the rate per 10 000 person-years of assault by firearm discharge was 2.5 with an additional 1.5 deaths from firearm discharge with undetermined intent. Deaths from drowning and submersion were 1.3/10 000 person-years. Intentional self-harm deaths included self-poisoning and hanging–

**Table 2** Mortality by baseline health-risk behaviours and states, social attributes and history of injury, Thai Cohort Study

| Social and health attributes | Vital status by attributes, per cent | | | | | Incidence/10 000 person-years | |
| --- | --- | --- | --- | --- | --- | --- | --- |
| | | | | Injury deaths (204) | | | |
| | Alive (86 457) | Other deaths (376) | Injury deaths (204) | Transport (84) | Non-transport (120) | Transport (84) | Non-transport (120) |
| Health-risk attributes | | | | | | | |
| Smoking | | | | | | | |
| Never | 72.3 | 51.1 | 57.5 | 66.3 | 51.3 | 9 | 10 |
| Current | 10.0 | 19.8 | 24.0 | 21.7 | 25.6 | 21 | 35 |
| Former | 15.8 | 26.1 | 15.5 | 9.6 | 19.7 | 6 | 17 |
| Alcohol drinking | | | | | | | |
| Never | 26.5 | 22.9 | 19.8 | 21.4 | 18.6 | 8 | 10 |
| Occasional | 59.8 | 49.5 | 60.9 | 61.9 | 60.2 | 10 | 14 |
| Regular | 4.8 | 9.7 | 7.9 | 9.5 | 6.8 | 19 | 19 |
| Stop | 8.9 | 18.0 | 11.4 | 7.1 | 14.4 | 8 | 22 |
| Ever drunk driving in past year | | | | | | | |
| Yes | 25.4 | 26.5 | 42.4 | 52.9 | 35.0 | 20 | 19 |
| Do not usually drive | 8.8 | 8.2 | 8.8 | 9.4 | 8.3 | 10 | 13 |
| Health and social attributes | | | | | | | |
| Self-assessed health | | | | | | | |
| Poor or very poor | 4.6 | 8.8 | 3.9 | 2.4 | 5.0 | 7 | 16 |
| Chronic conditions | | | | | | | |
| Yes | 12.5 | 29.3 | 10.8 | 13.1 | 9.2 | 10 | 13 |
| Doctor-diagnosed depression | | | | | | | |
| Yes | 3.4 | 5.9 | 6.9 | 4.8 | 8.3 | 13 | 34 |
| Social capital | | | | | | | |
| Low trust | 38.2 | 36.9 | 34.5 | 35.4 | 33.9 | 9 | 12 |
| Low support | 25.5 | 33.2 | 20.1 | 22.6 | 18.3 | 9 | 20 |
| Low interaction | 23.3 | 25.5 | 28.9 | 22.6 | 33.3 | 9 | 10 |
| Injury reported in 2005 | | | | | | | |
| Number of injuries | | | | | | | |
| At least once | 20.2 | 29.5 | 27.9 | 33.3 | 24.2 | 16 | 16 |
| Location of injury | | | | | | | |
| Home | 5.3 | 7.1 | 5.4 | 5.9 | 5.0 | 10 | 14 |
| Road | 5.9 | 4.8 | 11.7 | 16.5 | 8.3 | 25 | 20 |
| Work | 3.9 | 6.4 | 5.4 | 9.5 | 2.5 | 11 | 13 |

**Table 3** Injury mortality by ICD-10, Thai Cohort Study

| Types of injury deaths | Number of deaths | Rate per 10 000 person-years |
|---|---|---|
| Transport injuries | | |
| V01–V09 pedestrian | 2 | 0.2 |
| V20–V29 motorcycle rider | 14 | 1.6 |
| V40–V49 car occupant | 15 | 1.7 |
| V50–59 occupant of pick-up truck or van | 3 | 1.0 |
| V80–V89 other land transport accident | 9 | 0.1 |
| V89.2 person injured in unspecified motor vehicle | 39 | 4.5 |
| V90–V94 water transport | 1 | 0.1 |
| V95–V97 air and space transport | 1 | 0.1 |
| Non-transport injuries | | |
| W00–W19 falls | 2 | 0.3 |
| W65–W74 drowning and submersion | 10 | 1.3 |
| W75–W84 other threats to breathing | 1 | 0.1 |
| W87 exposure to electric current | 2 | 0.3 |
| X00–X09 exposure to smoke, fire and flames | 2 | 0.3 |
| X33 victim of lighting | 1 | 0.1 |
| X38 victim of flood | 1 | 0.1 |
| X58–X59 exposure to other unspecified factors | 16 | 2.1 |
| X60–X84 intentional self-harm | | |
| X65 intentional self-poisoning | 3 | 0.3 |
| X70 intentional self-harm by hanging, strangulation and suffocation | 7 | 0.8 |
| X85–Y09 assault | | |
| X95 assault by unspecified firearm discharge | 22 | 2.5 |
| X99 assault by sharp object | 3 | 0.3 |
| Y99 assault by other unspecified means | 5 | 0.6 |
| Y10–Y34 Event of undetermined intent | | |
| Y18 poisoning by and exposure to pesticides | 1 | 0.1 |
| Y20 hanging, strangulation and suffocation | 3 | 0.3 |
| Y22–Y24 firearm discharge, undetermined intent | 13 | 1.5 |
| Y25 contact with explosive material | 1 | 0.1 |
| Y28–Y29 contact with sharp of blunt object | 4 | 0.5 |
| Y34 unspecified event, undetermined intent | 23 | 2.6 |

ICD, International Classification of Diseases.

strangulation–suffocation with death rates of 0.3 and 0.8 per 10 000 person-years, respectively.

To examine predictors of injury deaths (table 4), we used multinomial logistic regression with three outcome categories: alive (reference), non-injury deaths and injury deaths (study outcome). Highlighted in bold were results that were statistically significant at p<0.05. In the first column of ORs, the results are adjusted for age and sex; in the second column the ORs are adjusted for all covariates. All ORs compare the odds of injury death with the odds of staying alive.

We first calculated age–sex AORs for each potential exposure and for covariates. In the age–sex-adjusted mode, being younger than 30 years, being male, residing in the south, currently smoking, drunk driving in the past year, ever having been diagnosed for depression, injury incidence in the year preceding baseline and reported road injury in 2005, were all associated with injury mortality.

We then proceed to multivariate analysis (table 4). After mutually adjusting for all tabulated covariates, statistically significant predictors of injury mortality were being male (AOR 3.87), residing in the southern areas (AOR 1.71), being a current smoker (AOR 1.56), history of drunk driving (AOR 1.49) and ever having been diagnosed for depression (AOR 1.91). Other covariates such as being young, having low social support and reporting road injury in the past year at baseline had predictive AORs ranging from 1.4 to 1.6, but these substantive estimates were not statistically significant for overall injury. Further investigation into types of injury (data not shown) revealed that younger age was a strong predictor for transport injury deaths (AOR 4.12, 95% 1.03 to 16.5) and low social support for non-transport injury deaths (AOR 1.64, 95% 1.04 to 2.59).

In marked contrast to the injury deaths, cohort members who died from other causes had different sets of statistically significant risks at 2005 baseline which included older age, residing in Bangkok, reporting poor self-assessed health and having metabolic or cardiovascular chronic conditions (data not shown). The only risk factor that non-injury deaths have in common with injury deaths was being a current smoker.

**Table 4** Age–sex adjusted and multivariate predictors of injury mortality, Thai Cohort Study

| | Multinomial* adjusted ORs (95% CI) | | | |
| --- | --- | --- | --- | --- |
| | Age–sex adjusted | | Multivariate† | |
| 2005 baseline covariates | Alive | Injury death | Alive | Injury death |
| Geodemographic covariates | | | | |
| Age groups in years 20–29 | Ref | **1.45 (1.06 to 1.99)** | Ref | 1.51 (0.99 to 2.32) |
| 30–39 | | | | 0.99 (0.54 to 1.83) |
| 40–49 | | 0.76 (0.45 to 1.28) | | 1.65 (0.61 to 4.43) |
| ≥50 | | 1.29 (0.59 to 2.84) | | |
| Sex (female) | Ref | | Ref | |
| Male | | **3.69 (2.69 to 5.07)** | | **3.87(2.39 to 6.26)** |
| Marital status (married) | Ref | | Ref | |
| Not married | | 1.09 (0.77 to 1.56) | | 1.11 (0.73 to 1.69) |
| Divorced/widowed | | 1.55 (0.80 to 3.02) | | 1.47 (0.69 to 3.14) |
| Personal monthly income | | | | |
| ≤3000 Baht | | 1.24 (0.80 to 1.93) | | 1.10 (0.48 to 1.53) |
| 3001–7000 | | 1.25 (0.89 to 1.75) | | 1.15 (0.78 to 1.69) |
| 7001–20 000 | | 0.81 (0.46 to 1.43) | | 0.83 (0.43 to 1.61) |
| >20 000 | Ref | | Ref | |
| Regions (*Central/east*) | Ref | | Ref | |
| Bangkok | | 0.85 (0.51 to 1.41) | | 0.86 (0.48 to 1.53) |
| North | | 1.38 (0.92 to 2.08) | | 1.27 (0.80 to 2.01) |
| Northeast | | 1.29 (0.87 to 1.93) | | 0.98 (0.61 to 1.56) |
| South | | **1.96 (1.30 to 2.97)** | | **1.71 (1.05 to 2.79)** |
| Lifecourse residence (rural–rural) | Ref | | Ref | |
| Urban–rural | | 0.91 (0.66 to 1.27) | | 0.97 (0.65 to 1.44) |
| Rural–urban | | 1.03 (0.52 to 2.04) | | 1.32 (0.73 to 1.44) |
| Urban–urban | | 0.87 (0.59 to 1.29) | | 1.12 (0.71 to 1.77) |
| Health and social covariates | | | | |
| Smoking (never) | Ref | | Ref | |
| Current | | **1.70 (1.07 to 2.05)** | | **1.56 (1.03 to 2.37)** |
| Former | | 0.77 (0.45 to 1.32) | | 0.74 (0.47 to 1.18) |
| Alcohol drinking (never) | Ref | | Ref | |
| Occasional | | 0.87 (0.60 to 1.27) | | 0.40 (0.12 to 1.30) |
| Regular | | 1.06 (0.58 to 1.95) | | 0.38 (0.10 to 1.40) |
| Stop | | 1.08 (0.64 to 1.84) | | 0.63 (0.18 to 2.15) |
| Drink driving past year (never) | Ref | | Ref | |
| Yes | | **1.50 (1.07 to 2.12)** | | **1.49 (1.01 to 2.20)** |
| Self-assessed health (good) | Ref | | Ref | |
| Poor or very poor | | 1.09 (0.58 to 2.07) | | 0.75 (0.32 to 1.71) |
| Depression (no) | Ref | | Ref | |
| Yes | | **2.15 (1.25 to 3.71)** | | **1.91 (1.00 to 3.69)** |
| Chronic illness (no) | Ref | | Ref | |
| Yes | | 0.80 (0.50 to 1.2) | | 0.84 (0.50 to 1.44) |
| Social capital | | | | |
| Low social trust (ref not low) | | 0.84 (0.60 to 1.19) | | 0.90 (0.76 to 1.07) |
| Low social support (ref not low) | | 1.29 (0.58 to 2.84) | | 1.37 (0.96 to 1.96) |
| Low social interaction (ref not low) | | 0.86 (0.64 to 1.15) | | 0.77 (0.50 to 1.20) |
| Injury reported in 2005 | | | | |
| Injuries in the past year (no) | Ref | | Ref | |
| At least once | | **1.41 (1.04 to 1.92)** | | 1.12 (0.70 to 1.82) |
| Location of injury | | | | |
| Home (ref no) | | 1.13 (0.62 to 2.08) | | 1.19 (0.56 to 2.55) |
| Road (ref no) | | **1.81 (1.17 to 2.79)** | | 1.58 (0.87 to 2.84) |
| Work (ref no) | | 1.22 (0.66 to 2.25) | | 1.15 (0.54 to 2.44) |

*Multinomial logistic regression compares the odds of injury deaths to the odds of remaining alive by predictor covariate category values, after adjusting for age–sex or other covariates.
†Mutually adjusted for all predictor covariates presented in this table.
Results in bold typeface were significant at p<0.05.

## DISCUSSION

This study is embedded in an overarching investigation of health-risk transition in Thailand, where similar to 'other' middle-income Southeast Asian countries, injury has been and still is a major population health burden. We make use of our large cohort of adults to investigate risk factors associated with injury mortality, linking deaths over 5 years (2005–2010) to individual characteristics and social and health attributes recorded at baseline (2005). The results revealed that injuries constituted almost one-third of all deaths in the cohort, and some 40% of those were from transport and nearly 60% were non-transport injuries. Connecting to baseline information provided 5 years earlier, we identified epidemiologically and statistically significant predictors of injury mortality. These predictors included certain geo-demographic characteristics (male sex, southern residents), health-risk behaviours (smoking and history of drunk driving) and adverse health diagnoses (depression).

Our injury mortality findings provide further information adding to our previous knowledge on risk factors associated with injury morbidity in the cohort, especially being male, having a history of depression and drunk driving.[18–20] The effect of smoking on injury was also shown in other studies after controlling for covariates including alcohol drinking. Plausible explanations include accidental fire hazard and distraction or inattention for road traffic hazards.[25 26] Non-transport injury deaths presented here reflected the current political situation with violence in the southern part of Thailand since 2004.[28 29] For covariates with substantial (but not statistically significant) point estimates of overall injury death (young age, low social support, history of road injury), AORs ranged from 1.4 to 1.6.

Our results support other published international information on risk factors related to injury mortality, showing young men and drunk drivers at high risk of road crash death.[30 31] A systematic review on alcohol consumption and collision risk concluded that there is no safe level of drinking and even less than two drinks per occasion can almost double the odds of most types of injury.[30]

Our study also found social capital to be protective against injury mortality, supporting previous research.[32–34] Indeed, we found low social support to be a predictor of non-transport injury mortality. But social support did not relate to transport injury mortality. Despite the relatively small number of deaths in our study, we found that low social support was particularly associated with intentional self-harm and assault but these associations were not statistically significant.

The advantage of our study is its large participation by Thai adults who completed a comprehensive 20-page baseline questionnaire in 2005, reporting on a wide array of social and health characteristics. This provided a platform for investigation of causal pathways and elimination of many confounders. While our cohort members share similar distribution of sex, modest income and geographical residence with the general Thai population, they also completed high school education which facilitated their ability to respond to our detailed questionnaire. We also noted that a few (<5%) cohort members were 'missing' data for variables used in various models but this problem was not numerically significant. We acknowledge that our study captured the 5-year mortality rate after the baseline questionnaire in 2005 and the number of deaths was relatively small. However, the citizen IDs provided by cohort members at the baseline can be used to monitor patterns of mortality into the future and eventually a full account of cohort mortality will be possible.

Our research contributes to limited longitudinal evidence linking risk factors to injury death and is one of the first studies in middle-income Southeast Asia. We have achieved our aim of identifying vulnerable population subgroups at risk of injury deaths. As well, we complemented routinely collected administration death registration with a longitudinal health assessment providing information on individual, social and health history factors. This information is invaluable in yielding insight into certain risk traits such as being a young male, reporting and having a medical diagnosis of depression, which could inform injury prevention policies and strategies suitable for transitional countries with limited resources and changing patterns of mortality.

### Author affiliations

[1]National Centre for Epidemiology and Population Health, The Australian National University, Canberra, Australian Capital Territory, Australia
[2]Monash Injury Research Institute, Monash University, Melbourne, Australia
[3]Genetics and Population Health Division, QIMR Berghofer Medical Research Institute, Brisbane, Queensland, Australia
[4]School of Human Ecology, Sukhothai Thammathirat Open University, Nonthaburi, Thailand

**Acknowledgements** The authors thank the staff at Sukhothai Thammathirat Open University (STOU) who assisted with student contact and the STOU students who are participating in the cohort study. The authors also thank Dr Bandit Thinkamrop and his team from Khon Kaen University for support on initial data processing. As well we would like thank the Thai Ministry of Interior and Ministry of Public Health for mortality investigation and cause of death data linkage. Matthew Kelly and Peter Sbirakos provided language editorial support.

**Collaborators** The Thai Cohort Study Team, Thailand: Jaruwan Chokhanapitak, Suttanit Hounthasarn, Suwanee Khamman, Daoruang Pandee, Suttinan Pangsap, Tippawan Prapamontol, Janya Puengson, Sam-ang Seubsman, Boonchai Somboonsook, Nintita Sripaiboonkij, Pathumvadee Somsamai, Duangkae Vilainerun, Wanee Wimonwattanaphan, Tewarit Somkotra, Benjawan Tawatsupa, Wimalin Rimpeekool. Australia: Chris Bain, Emily Banks, Cathy Banwell, Bruce Caldwell, Gordon Carmichael, Tarie Dellora, Jane Dixon, Sharon Friel, David Harley, Matthew Kelly, Tord Kjellstrom, Lynette Lim, Anthony McMichael, Tanya Mark, Penny Haora, Adrian Sleigh, Lyndall Strazdins, Susan Jordan, Roderick McClure, Janneke Berecki-Gisolf, Vasoontara Yiengprugsawan.

**Contributors** VY conceptualised, analysed and drafted the manuscript. JB-G, CB, RM provided inputs to improve the draft. SS and AS designed and executed the Thai Cohort Study. AS provided comments and guidance on revisions. All authors read and approved the final manuscript.

**Funding** This study was supported by the International Collaborative Research Grants Scheme with joint grants from the Wellcome Trust UK (GR071587MA) and the Australian National Health and Medical Research Council (268055), and as a global health grant from the NHMRC (585426).

**Competing interests** None.

**Patient consent** Obtained.

**Ethics approval** Informed written consent was obtained from all participants. All students were advised that they could withdraw, or not participate, without any effect on their academic progress. The questionnaires never sought sensitive personal information and no biological samples were taken. Ethics approval was obtained from Sukhothai Thammathirat Open University Research and Development Institute (protocol 0522/10) and the Australian National University Human Research Ethics Committee (protocols 2004/344 and 2009/570).

**Provenance and peer review** Not commissioned; externally peer reviewed.

**Data sharing statement** Data are available through a data access agreement which includes guarantees regarding ethical conduct and scientific quality of any proposed analyses and publications. Anyone wanting access should contact ACS or SS (Thai Cohort Study Principal Investigators).

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
