## [Reviewer comments · BMJ Open]

Some articles will have been accepted based in part or entirely on reviews undertaken for other BMJ Group journals. These will be reproduced where possible.

ARTICLE DETAILS

TITLE (PROVISIONAL)	Predictors of injury mortality: findings from a large national cohort in Thailand
AUTHORS	Yiengprugsawan, Vasoontara; Berecki-Gisolf, Janneke; Bain, Christopher; McClure, Roderick; Seubsman, Sam-ang; Sleigh, Adrian

VERSION 1 - REVIEW

REVIEWER	Guoqing Hu Central South University, China
REVIEW RETURNED	01-Mar-2014

GENERAL COMMENTS	This paper reported some interesting predictors of injury mortality based on a national cohort. The information is valuable for injury prevention of Thailand. However, I am not sure whether these findings could add to injury prevention knowledge. Please see specific comments below. 1. The selection bias of study subjects should be seriously revisited since they were recruited from distance-learning students. From age distribution in Table 1, we could easily find the sample doesn't look representative. If this is true, the data should be analyzed by including the weight.2. The introduction did not present a specific research question. I cannot read any research assumptions on any predictors of injury mortality although the authors aimed to explore the predictors. Especially, I did not read any justification for the selection of predictors like connected to depression. The current version did not clearly describe the potential contribution to injury prevention knowledge.3. Clear research questions always determine the sample size of cohort. However, I did not read this information and am not sure whether the sample is sufficient to detect the relationship of potential predictors and injury deaths.4. The quality of national death records may affect the determination of whether a student lived or died when the follow-up ended since the authors used matching the baseline database and national death record.5. It is hard for me to understand the selection of smoking as a potential predictor. Is there any underlying mechanism affecting the relationship?6. The limitation should include the potential influence of selection bias.
--

	7. The grading of social and health attributes should be given clearly definitions.
--	---

REVIEWER	Etienne Pracht University of South Florida, USA
REVIEW RETURNED	10-Mar-2014

GENERAL COMMENTS	I marked "no" concerning research ethics given the presence of a Citizen's ID. The authors should address briefly how the data is being safeguarded or if this is even a concern in Thailand. Trivial: In results section, change "history of drink driving" to "history of drunk driving" or "history of drinking and driving." On page 3, under "Key Messages" change "drink driving" to "drunk driving" or "drinking and driving." (same on page 6, 2nd paragraph; page 7, 1st paragraph, page 8, 1st and 2nd paragraph; page 9 2nd paragraph) On page 4, 1st paragraph, remove "of" before "50 million." On page 5, 1st paragraph: use of the first person "we" is unconventional. Consider rewriting the sentence and removing the first person reference. On page 9, 2nd paragraph, change "Our study also ... support to be predictor of on-transport injury ..." to "Our study also ... support to be predictive of on-transport injury ..." Suggestions: Particularly for transport related mortality, use CID10 codes to classify types of injury, for example traumatic brain injury (TBI), skull and spinal cord injury (SSCI) other than TBI, fractures other than TBI or SSCI, injuries to the thorax, and vascular injuries. This will provide more insight concerning the mechanism and causes of injury and may be useful for policy purposes. High rates of TBI and SSCI for example may link directly to inadequate use of seatbelts or helmets in case of motor cycles.
--

REVIEWER	Torsten Eken, MD PhD, senior researcher / consultant anaesthesiologist Dept. of Anaesthesiology Oslo University Hospital, Ullevål Oslo Norway
REVIEW RETURNED	12-Mar-2014

GENERAL COMMENTS	This manuscript is a prospective study from Thailand derived from a research cohort of 87,037 distance-learning students, aiming to
---

identify predictors for injury mortality during the first five years after baseline. The data are highly interesting, especially because they are collected from a large and geographically dispersed population in a middle-income Southeast Asian country which conceivably has other challenges and risk factors than high-income Western countries, where most such studies have been undertaken. It will be even more interesting to follow future studies with longer observation times of the same cohort.

General comment:

Language in the manuscript is of somewhat variable quality, see e.g. Introduction (“an estimated of 50 million”; mixed past and present tense), Methods (“doctored diagnosed diseases”; “analysis of deaths into the future”), and Table 2 (category “Not usually drive”, category group “Number of injury”). The manuscript would benefit from a thorough revision by a native English speaker.

Specific comments:

The “Article summary” section (Page 3) should be prepared according to Instructions for authors, with the heading 'Strengths and limitations of this study' and up to five bullet points that relate specifically to the study reported.

The manuscript would benefit from explicitly reporting according to the STROBE Statement. In particular, bias and missing data do not seem to be addressed in sufficient detail. What was the response rate for the initial questionnaire?

The Results section suffers from a high degree of repetition of data already presented in the tables (e.g. Page 8, lines 7-20).

In Results, first paragraph (Page 6, lines 52-54), it is stated that about half of the cohort members at baseline were less than 30 years old. According to Table 1, this age group constitutes less than a third of the cohort. In the same sentence it is stated that 30% lived in Bangkok, while Table 1 shows this fraction to be less than 18%.

Categories and groups in Table 2 and Table 4 are labelled differently. This should be avoided. In particular, it is confusing when “Social support” with “no” as reference category actually means “Low social support” (Table 4). Additionally, only one of the terms “geo-demographic” (text and Table 2) and “demo-geographic” (Table 4) should be used in the manuscript.

In the first Discussion paragraph (Page 8, lines 45-47), the percentages of transport and non-transport injury deaths (41% and 59%) should not be rounded to only one significant digit.

Analysis results should not be introduced for the first time in Discussion and without showing data (Page 9, lines 11-15; Page 9, lines 30-31), neither should these undocumented results be included as a key message (Page 3, lines 19-20).

Avoid the use of “social support” as a synonym for “low social support” (Page 9, line 26; cf. above comment regarding Table 4).

The number of participants in the cohort is impressive, but the question whether the cohort is representative for the general

VERSION 1 – AUTHOR RESPONSE

REVIEWER: Guoqing Hu

Institution and Country: Central South University, China

1. The selection bias of study subjects should be seriously revisited since they were recruited from distance-learning students. From age distribution in Table 1, we could easily find the sample doesn't look representative. If this true, the data should be analyzed by including the weight.

>>>>>>

Thank you for your comments. The aim of the study is to investigate predictors of injury mortality as part of the Thai Cohort Study. We agree that the cohort is younger than the Thai population; but we note that our cohort was not designed to measure relative prevalence. Indeed, the utility of prospective longitudinal cohort studies derives from the robust estimates produced for relative risks. These relative risk estimates are based on incidence and do not change if the cohort is reweighted (to simulate the population) provided the weighting variables (such as age and sex) are included in the multivariable model estimating the relative risks (Mealing et al 2010 BMC Medical Research Methodology (biomedcentral.com/1471-2288/10/26/))

For our analyses, we included the weighting variables as confounders in the final model. It is worth noting other epidemiological literature (Miettinen OS: Theoretical epidemiology principles of occurrence research in medicine. New York: Wiley; 1985) dealing with cohorts designed to estimate relative risks are not usually weighted to the source population. The validity of the relative risk measured is inherent in cohort design in both theory and practice and we would be reluctant to depart from this standard practice.

>>>>>>

2. The introduction did not present a specific research question. I cannot read any research assumptions on any predictors of injury mortality although the authors aimed to explore the predictors. Especially, I did not read any justification for the selection of predictors like connected to depression. The current version did not clearly describe the potential contribution to injury prevention knowledge.

>>>>>>

We have revised the last paragraph of the Introduction section as follows: "Informed by our earlier research on injury morbidity, and by related published information, this study investigated injury in more depth using mortality as the outcome. Our study linked cohort outcomes (survival, non-injury death, injury death) to an array of exposures recorded at baseline including geo-demographic attributes, social covariates, health and psychological states, and health-risk behaviours. This analysis is both prospective and cohort-based and fills an important gap regarding our knowledge of injury risks as an emerging public health problem in a middle-income Asian country

going through the health-risk transition."

>>>>>>

3. Clear research questions always determine the sample size of cohort. However, I did not read this information and am not sure whether the sample is sufficient to detect the relationship of potential predictors and injury deaths.

>>>>>>

Overall, the cohort was designed to answer many public health questions in transitional Thailand and at the start we were aware of the utility of large cohorts. Examples include the US Nurses Health Study, UK BioBank, Million Women Study, and Australian 45 and Up. With the baseline sample (n=87037), our cohort is large enough to detect quite small relative effects. Indeed, we note that after five years we were already able to detect as significant Adjusted Odds Ratio as modest as 1.4 (please see results in Table 4 of the revised manuscript).

>>>>>>

4. The quality of national death records may affect the determination of whether a student lived or died when the follow-up ended since the authors using matching the baseline database and national death record.

>>>>>>

The Thai death data now have almost complete coverage so almost all deaths will be detected through this national mortality database. We have included additional information in the Methods section as follows: "The completeness of death registration in Thailand was 86% from 1950-2000 (ref 10), but over the last decade coverage improved to 95% (ref 23)."

>>>>>>

5. It is hard for me to understand the selection of smoking as a potential predictor. Is there any underlying mechanism affecting the relationship?

>>>>>>

We have now explained such relationships based on previous literature under Exposures and Confounders in the Methods section: "Health-risk behaviours included smoking and alcohol which were shown to be independently associated with injury risk"(refs 26-27)

As well as in the 2nd paragraph in the Discussion sections as follows:

"Effect of smoking on injury was also shown in other studies after controlling for covariates including alcohol. Plausible explanations include accidental fire hazard and distraction or inattention for road traffic hazards." (ref 26-27)

>>>>>>

6. The limitation should include the potential influence of selection bias.

>>>>>>

We have investigated our cohort for selection bias for a variety of scenarios and have not found any systematic link between missingness and morbidity outcomes of interest. For mortality analysis, by definition we follow-up the entire cohort because the data on deaths come from a different source which is virtually 100% complete (ie no missing data). So we are confident that the issue of missing specific data when modelling will not affect the results and from our earlier analyses will all be occurring at random. We have included additional information in the Methods section under Data processing and statistical analysis as follows: "Individuals with missing data for given analyses were excluded (<5%), so totals vary slightly according to available information."

And under limitation of Discussion: "We also noted that a few (<5%) cohort members were 'missing' data for variables used in various models but this problem was not numerically significant."

>>>>>>

7. The grading of social and health attributes should be given clearly definitions. We have now expanded the Methods section, including more definitions under Exposures and confounders in the Methods section.

REVIEWER: Etienne Pracht
Institution and Country: University of South Florida, USA

Given the presence of a Citizen's ID. The authors should address briefly how the data is being safeguarded or if this is even a concern in Thailand.

>>>>>>

Thank you for your comments, confidentiality is a concern in Thailand and Australia and was addressed in the ethical approval. We have now included additional information in the second paragraph of the Methods section as follows:

...These confidential ID numbers were safeguarded and stored at STOU in a secure office on the main campus with 24-hour guards on patrol. The working files of these data were de-identified and no individual information will be released or displayed in any format.

>>>>>>

Trivial:

In results section, change "history of drink driving" to "history of drunk driving" or "history of drinking and driving." On page 3, under "Key Messages" change "drink driving" to "drunk driving" or "drinking and driving." (same on page 6, 2nd paragraph; page 7, 1st paragraph, page 8, 1st and 2nd paragraph; page 9 2nd paragraph)

>>>>>>

We prefer to use our own 'idiolect' and that means 'drink driving'. If the reviewer has a strong view against this, we will change this in the next revision.

>>>>>

On page 4, 1st paragraph, remove "of" before "50 million."

>>>>>

Revised.

>>>>>

On page 5, 1st paragraph: use of the first person "we" is unconventional. Consider rewriting the sentence and removing the first person reference.

>>>>>

Revised.

>>>>>

On page 9, 2nd paragraph, change "Our study also ... support to be predictor of on-transport injury ..." to "Our study also ... support to be predictive of on-transport injury ..."

>>>>>

Revised.

>>>>>

Suggestions:

Particularly for transport related mortality, use CID10 codes to classify types of injury, for example traumatic brain injury (TBI), skull and spinal cord injury (SSCI) other than TBI, fractures other than TBI or SSCI, injuries to the thorax, and vascular injuries. This will provide more insight concerning the mechanism and causes of injury and may be useful for policy purposes. High rates of TBI and SSCI for example may link directly to inadequate use of seatbelts or helmets in case of motor cycles.

>>>>>

We agree with your suggestions. In view of 5-year injury mortality (n=204) examined here it was not statistically possible at this stage to investigate specific details of injury deaths. Our current data do not contain deaths from traumatic brain injury or skull and spinal cord injury. However, given a longer period of follow-up (feasible with Citizen ID-matched death records) allowing for sufficient injury deaths, it will be possible to investigate such injury mechanisms in the future.

>>>>>

REVIEWER: Torsten Eken

Institution and Country: Oslo University Hospital, Ullevål, Norway

Language in the manuscript is of somewhat variable quality, see e.g. Introduction ("an estimated of 50 million"; mixed past and present tense), Methods ("analysis of deaths into the future"), and Table 2 (category "Not usually drive", category group "Number of injury"). The manuscript would benefit from a thorough revision by a native English speaker.

>>>>>>

Thank you for your comments, we have sought assistance from a native English speaker to review and edit our draft manuscript.

>>>>>>

Specific comments:

The "Article summary" section (Page 3) should be prepared according to Instructions for authors, with the heading 'Strengths and limitations of this study' and up to five bullet points that relate specifically to the study reported.

>>>>>>

We have now revised the Article summary according to Instruction for authors.

>>>>>>

The manuscript would benefit from explicitly reporting according to the STROBE Statement. In particular, bias and missing data do not seem to be addressed in sufficient detail. What was the response rate for the initial questionnaire?

>>>>>>

We have revisited the STROBE Statement and have added information on bias and missing data in the Methods and Discussion sections as well as response rate for the initial questionnaire as suggested.

>>>>>>

The Results section suffers from a high degree of repetition of data already presented in the tables (e.g. Page 8, lines 7-20).

>>>>>>

Comments noted and have now shortened this section to highlight the key findings.

>>>>>>

In Results, first paragraph (Page 6, lines 52-54), it is stated that about half of the cohort members at baseline were less than 30 years old. According to Table 1, this age group constitutes less than a third of the cohort. In the same sentence it is stated that 30% lived in Bangkok, while Table 1 shows this fraction to be less than 18%.

>>>>>>

Thank you for pointing these out, we have amended these sections.

>>>>>>

Categories and groups in Table 2 and Table 4 are labelled differently. This should be avoided. In particular, it is confusing when "Social support" with "no" as reference category actually means "Low social support" (Table 4). Additionally, only one of the terms "geo-demographic" (text and Table 2) and "demo-geographic" (Table 4) should be used in the manuscript.

>>>>>>

We have corrected the social capital variables and also changed to 'geo-demographic' in the manuscript.

>>>>>

In the first Discussion paragraph (Page 8, lines 45-47), the percentages of transport and non-transport injury deaths (41% and 59%) should not be rounded to only one significant digit.

>>>>>

We have reworded this section to: ..some 40% of those were from transport and nearly 60% were non-transport injuries.

>>>>>

Analysis results should not be introduced for the first time in Discussion and without showing data (Page 9, lines 11-15; Page 9, lines 30-31), neither should these undocumented results be included as a key message (Page 3, lines 19-20).

>>>>>

We have now moved this section as additional information in Results. We also have removed this information from Key Message section.

>>>>>

Avoid the use of "social support" as a synonym for "low social support" (Page 9, line 26; cf. above comment regarding Table 4).

>>>>>

Thank you, we have clarified this.

>>>>>

The number of participants in the cohort is impressive, but the question whether the cohort is representative for the general population is not sufficiently addressed (Page 9, lines 41-46).

>>>>>

We have now expanded this section as suggested.

>>>>>

VERSION 2 – REVIEW

REVIEWER	Torsten Eken Oslo University Hospital Ullevål, Norway
REVIEW RETURNED	05-May-2014

GENERAL COMMENTS	The manuscript has improved substantially, and I have only a small number of minor comments: - The "Article summary" section (Page 3) still contains three headings and eight bullet points. This section should be prepared according to Instructions for authors: One single heading 'Strengths and limitations of this study' followed by up to five bullet points in total.
--

	- Methods, Page 6, Line 32: "70001" should probably be "7001".- Results, Page 8, Line 34: Having low social support does not have a statistically significant association with injury mortality (Table 4: AOR 1.29 [0.58-2.84]).- Table 4, Page 17, Line 48: "084" should probably be "0.84".
--	---

VERSION 2 – AUTHOR RESPONSE

Reviewer (Dr. Torsten Eken),

Thank you very much for re-reviewing manuscript. With apologies for the inconvenience with our marked copy, in this revision we have used track changes documenting changes as well as deletions. We also note your suggestion for future manuscripts.

We have also addressed four additional comments as follows:

- The "Article summary" section (Page 3) still contains three headings and eight bullet points. This section should be prepared according to Instructions for authors: One single heading 'Strengths and limitations of this study' followed by up to five bullet points in total. We have now revised this section under 'Strengths and limitations of this study' with four points.

- Methods, Page 6, Line 32: "70001" should probably be "7001". We have corrected this.

- Results, Page 8, Line 34: Having low social support does not have a statistically significant association with injury mortality (Table 4: AOR 1.29 [0.58-2.84]). Thank you, we have amended this.

- Table 4, Page 17, Line 48: "084" should probably be "0.84". Corrected.

We have also made some small changes in the manuscript (eg deletions of abbreviation which was not subsequently referred to).